# Shaping care home COVID-19 testing policy: a protocol for a pragmatic cluster randomised controlled trial of asymptomatic testing compared with standard care in care home staff (VIVALDI-CT)

Natalie Adams [1], Oliver Stirrup,[2] James Blackstone [3], Maria Krutikov,[1] Jackie A Cassell [4,5] Dorina Cadar,[4,6] Catherine Henderson,[7] Martin Knapp,[7] Lara Goscé,[2,8] Ruth Leiser,[9] Martyn Regan,[5,10] Iona Cullen-Stephenson,[3] Robert Fenner,[3] Arpana Verma,[11] Adam Gordon [12,13] Susan Hopkins,[5] Andrew Copas [2] Nick Freemantle,[3] Paul Flowers [9] Laura Shallcross[1]

For numbered affiliations see end of article.

**Correspondence to**
Dr Laura Shallcross;
l.shallcross@ucl.ac.uk

## ABSTRACT

**Introduction** Care home residents have experienced significant morbidity, mortality and disruption following outbreaks of SARS-CoV-2. Regular SARS-CoV-2 testing of care home staff was introduced to reduce transmission of infection, but it is unclear whether this remains beneficial. This trial aims to investigate whether use of regular asymptomatic staff testing, alongside funding to reimburse sick pay for those who test positive and meet costs of employing agency staff, is a feasible and effective strategy to reduce COVID-19 impact in care homes.

**Methods and analysis** The VIVALDI-Clinical Trial is a multicentre, open-label, cluster randomised controlled, phase III/IV superiority trial in up to 280 residential and/or nursing homes in England providing care to adults aged >65 years. All regular and agency staff will be enrolled, excepting those who opt out. Homes will be randomised to the intervention arm (twice weekly asymptomatic staff testing for SARS-CoV-2) or the control arm (current national testing guidance). Staff who test positive for SARS-CoV-2 will self-isolate and receive sick pay. Care providers will be reimbursed for costs associated with employing temporary staff to backfill for absence arising directly from the trial.

The trial will be delivered by a multidisciplinary research team through a series of five work packages. The primary outcome is the incidence of COVID-19-related hospital admissions in residents. Secondary outcomes include the number and duration of outbreaks and home closures. Health economic and modelling analyses will investigate the cost-effectiveness and cost consequences of the testing intervention. A process evaluation using qualitative interviews will be conducted to understand intervention roll out and identify areas for optimisation to inform future intervention scale-up, should the testing approach prove effective and cost-effective. Stakeholder engagement will be undertaken to enable the sector to plan for results and their implications and

## STRENGTHS AND LIMITATIONS OF THIS STUDY

⇒ First trial to evaluate the benefits and harms of regularly testing care home staff for COVID-19 to protect residents from severe outcomes following infection.
⇒ Process evaluation, economic and modelling analyses will provide insights into intervention feasibility and costs/cost-effectiveness, informing future public health policy.
⇒ The study demonstrates the potential for large-scale trials in care homes that are delivered in partnership with care providers and capitalise on routinely collected data.
⇒ The trial is being delivered in a rapidly changing policy and epidemiological context, which could undermine effective trial delivery.

to coproduce recommendations on the use of testing for policy-makers.

**Ethics and dissemination** The study has been approved by the London—Bromley Research Ethics Committee (reference number 22/LO/0846) and the Health Research Authority (22/CAG/0165). The results of the trial will be disseminated regardless of the direction of effect. The publication of the results will comply with a trial-specific publication policy and will include submission to open access journals. A lay summary of the results will also be produced to disseminate the results to participants.

**Trial registration number** ISRCTN13296529.

## INTRODUCTION
### Context
In England, approximately 380 000 people (4% of >65 years) live in 11 000 care homes for older adults. Most care home residents ('residents') are older than 85 years, at

least two-thirds live with dementia, and over half die within 12 months of admission to a care home.[1 2] Care home residents worldwide have experienced among the highest rates of COVID-19 mortality and morbidity,[3] and in England, they have also been subject to particularly strict and lengthy lockdown measures. Prolonged use of COVID-19 restrictions (eg, social isolation, visitor restrictions) has had a devastating impact on residents' well-being, and their physical and mental health, for example, depriving them of contact with family members in their final weeks of life.[4]

## Current knowledge

Public health disease control measures were deployed rapidly and simultaneously in care homes early in the pandemic to reduce infection spread, limiting any assessment of the impact of individual measures. There have been no interventional studies of non-pharmaceutical control measures to reduce COVID-19 infection in care homes, but a Cochrane rapid review (published in September 2021) identified 11 observational and 11 modelling studies, all from high-income countries.[5] The review grouped interventions into entry regulations (eg, reducing visitors), contact regulating and transmission-reducing measures (eg, personal protective equipment), surveillance (symptomatic and asymptomatic testing) and outbreak control measures. Across these domains the quality of evidence was poor. In addition, there was widespread recognition that some of these measures, such as preventing visitors from entering the care home, were associated with significant harm.

Throughout the pandemic, testing has been used in three ways to reduce transmission of infection: (1) symptomatic testing, (2) testing during outbreaks to reduce their duration and severity and (3) regular, asymptomatic testing.

In the UK, compliance with regular testing may have been driven by national policies incentivising testing, including with financial support (eg, Adult Social Care Rapid Testing Fund introduced in January 2021, Infection Control Fund introduced May 2020).[6 7] However, relatively few published studies have examined how these influence compliance with asymptomatic testing in care homes. We conducted a rapid systematic review, spanning January 2020 to July 2022. It highlighted 14 international papers,[8–21] published in English. No studies used an experimental design, and none reported, or evaluated, interventions designed to improve compliance with SARS-CoV-2 testing. The papers used a range of designs (eg, qualitative, cross-sectional quantitative, consensus building). Together these studies highlight the multi-levelled factors that have shaped adherence with SARS-CoV-2 testing in care homes. We then used the behaviour change wheel[22] as an approach to develop systematically potentially useful intervention content from the factors influencing testing identified within the literature. Subsequently, through a series of stakeholder engagement events with diverse care home staff and representatives

from the care home sector, we agreed the content of a multilevel intervention designed to maintain compliance with twice weekly lateral flow device (LFD) testing for COVID-19 within intervention care homes ('Test to Care').

There remains a lack of evidence on whether the benefits of regular testing for COVID-19 outweigh its harms, and if so, under which scenarios. There have been no attempts in the literature to consolidate the considerable expertise and learning on how to ensure compliance with testing in this setting. From a policy perspective, the key question remains over appropriate thresholds for turning testing 'on' and 'off' in response to varying levels of 'COVID-19 threat' (eg, high/low levels of infection in the community; the emergence of novel COVID-19 variants).

We posit that the best approach to address these questions is through a randomised clinical trial. Randomisation overcomes the problem of substantial heterogeneity among care homes as they vary in resident population, care provision and uptake of control measures such as vaccination in staff or use of facemasks, which limit the conclusions that can be drawn from observational studies. Although there are significant challenges associated with undertaking a trial in care homes in a changing policy and epidemiological context, there is an urgent need for high-quality evidence to inform the future use of testing for SARS-CoV-2 and potentially other infections in this setting.

## Study aims

We will investigate whether continued use of regular asymptomatic testing in staff is a feasible, effective and cost-effective strategy to reduce the impact of COVID-19 in care homes. Findings will inform testing policy across the UK for COVID-19 and add to knowledge on the use of testing in care homes to prevent other respiratory viruses, such as influenza. These objectives will be delivered through a series of five interlinked work packages (WPs), which are described in detail in online supplemental file 1.

## METHODS AND ANALYSIS
### Study design

VIVALDI-Clinical Trial (VIVALDI-CT) is a multicentre, open-label, cluster randomised controlled, phase III/IV superiority trial.

Each eligible care home will be randomised to either standard care (SARS-CoV-2 testing policy for care home staff that is in place nationally at the time of trial operation), or regular asymptomatic testing of care home staff for COVID-19 using LFDs combined with support payments for sickness absence and agency staff backfill.

### Study setting

VIVALDI-CT will take place in up to 280 residential and/or nursing homes in England providing care to adults aged >65 years.

## Recruitment

Due to rapid timescales for trial delivery, and the need to streamline and centralise data collection, we will primarily partner with providers that manage multiple care homes. We will first contact the senior management teams of providers that we have previously worked with in the Vivaldi study[23] to determine if they are interested in trial participation. Providers will be asked to supply a list of eligible care homes and confirm that the care home manager has provided consent for each listed home to participate. If we are unable to recruit sufficient homes from the Vivaldi network, we will work with provider representative organisations (eg, National Care Forum, Care England, National Care Association) to identify other eligible providers.

Homes will be selected to capture diversity in care home size, population (nursing/residential/dementia care), ethnicity, geographical location, rural/urban and provider type (for-profit/not-for-profit). Inclusive participation will be a focus by ensuring larger and smaller care groups are included in trial, as well as focusing on diverse settings.

## Inclusion and exclusion criteria

Only care home staff are eligible to participate in the testing intervention. This includes temporary (agency) staff with no restrictions, that is, catering, administrative and maintenance staff, in addition to those in a resident-facing role. However, all care home staff, as well as residents, visitors and relatives, are eligible to participate in interviews undertaken as part of the trial's process evaluation. All care home residents at participating homes are eligible for data collection and analysis of the outcomes specified.

Visitors, residents and relatives are not eligible to take part in the testing intervention. Staff who visit the care home to provide care but are not employed by the care home, for example, General Practitioners (GPs), health visitors, are not eligible to take part in either the interviews or the testing intervention.

## Primary outcome

The primary outcome is the incidence of COVID-19-related hospital admissions in residents defined as admissions with a relevant International Statistical Classification of Diseases and Related Health Problems 10th revision (ICD-10) code (COVID-19 hospitalisations to be defined as any hospital admission record with a primary or secondary ICD-10 code of 'U071') and/or admissions in residents who test positive for COVID-19 within 24 hours following admission or in the 7 days before hospital admission. This is considered the most important outcome for policy-makers.

## Secondary outcomes

Although we have adopted a healthcare/National Health Service (NHS) perspective for the primary outcome, we recognise the importance of capturing outcomes that are relevant to the social care sector, such as outbreaks and care home closures. This is reflected in our choice of secondary outcomes:

► Incidence rate of hospital admissions (all-cause) in residents for non-elective care measured as events per 100 000 person-days of follow-up over the duration of the trial.
► Incidence rate of COVID-associated mortality in residents measured as events per 100 000 person-days of follow-up over the duration of the trial.
► Incidence of all-cause mortality in residents measured as events per 100 000 person-days of follow-up over the duration of the trial.
► Testing uptake in staff measured as proportion of staff at each home participating in testing during each week of the trial.
► Prevalence of SARS-CoV-2 among staff who test measured as proportion of staff with positive test result among those with at least one test recorded during each week of the trial.
► Incidence rate of SARS-CoV-2 infections detected in residents measured as events per 100 000 person-days of follow-up over the duration of the trial.
► Incidence rate of home-level outbreaks measured as events per 1000 days of follow-up over the trial duration.
► Duration of outbreaks measured as days from first to last case within outbreaks occurring within the trial period.
► Incidence rate of care home closures due to outbreaks measured as events per 1000 days of follow-up over the duration of the trial.
► Proportion of staff per home who are off sick during each week of the trial.
► Proportion of all shifts filled by agency staff at each home each week.
► Costs per test.
► Testing metrics, for example, staff time taken to conduct the test at work.
► The impact of testing on resident, staff and visitors, for example, social care-related quality of life collected via interviews in WP3.

## Sample size

Based on observational data from the VIVALDI study, we found that over the 3-month period of January–March 2022, 1.8% of residents had a COVID-19-related hospital admission, and the intracluster correlation (ICC) across homes was 0.003 (95% CI 0.000 to 0.007). We assume that we will observe a cumulative incidence of around 3.0% in the trial, which would require a trial duration of 5–6 months if the incidence rate is similar to that in winter 2021/2022, in combination with a conservative ICC value of up to 0.01 (higher in line with the higher cumulative incidence compared with 3 months), and an average care home size of 35 residents with coefficient of variation in size of 0.5. With a total of 280 homes randomised 1:1 to trial arms and taking the usual two-sided test at

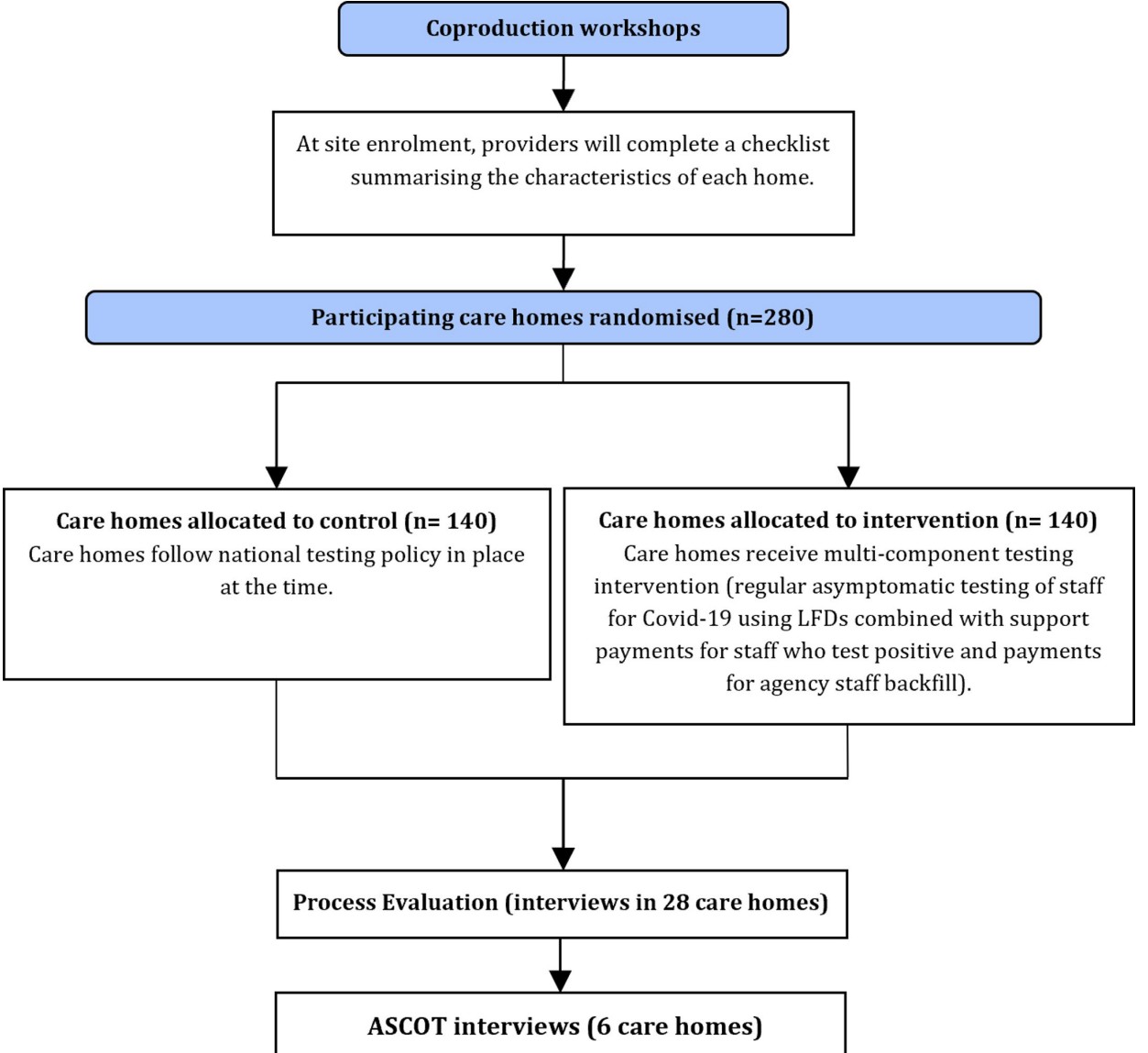

**Figure 1** Flow diagram of trial pathway. LFDs, lateral flow devices; ASCOT, Adult Social Care Outcomes Toolkit. Adult Social Care Outcomes Toolkit (ASCOT) interviews used to assess the impact of the intervention and outbreaks on residents.

5% significance level, the design provides 84% power to detect a reduction in COVID-19-related admissions due to intervention to 1.9% (relative risk 0.63).

### Timeline
The trial programme will run from November 2022 to April 2024. The recruitment of care home providers and operation of the intervention will take place between December 2022 and March 2023, with the possibility that this will be extended if deemed necessary for data collection. Online supplemental file 2 shows a participant timeline and figure 1 is a schematic of the trial pathway.

### Intervention allocation
Care homes will be randomised on a 1:1 ratio. If all providers are ready for trial participation at the same time, then all participating homes will be randomised at the same time. Otherwise, the homes from different providers will be randomised in a phased approach, as

they become ready. Randomisation will be performed by the trial statistician based on pseudorandom number generation after trial enrolment and before intervention implementation. Restricted randomisation (specifically covariate constrained randomisation) will be used to ensure balance on care home provider, size and region.

### Blinding
Researchers and staff of participating care homes will not be blinded to their intervention allocation, as this would not be feasible.

### Data collection
To facilitate trial setup and minimise the burden on care home staff, much of the data for analysis will be obtained from routinely collected healthcare information held within the UK COVID-19 Datastore.[24] This will include results of LFD and PCR tests for SARS-CoV-2 for staff and residents, information on hospital admissions and deaths

for residents, and vaccination history for residents. Data within the COVID-19 Datastore are linked to a pseudonymised ID at the level of each individual, which can be linked to care quality commission-ID (CQC-ID), a unique ID number provided by the CQC to identify each care home, for participating care homes and associated staff or resident status. A new study-specific pseudonymised ID will be created for each individual before export of data to University College London (UCL), in order to prevent any theoretical possibility of reidentification.

Hospital admissions data are linked to ICD-10 diagnostic codes (including COVID-19 codes); however, there is a lag of several months in the assignment of these codes. To allow timely monitoring of data quality for the primary outcome and limit the risk of omitting or double-counting hospital admissions in residents, during the intervention period providers will be asked to upload weekly lists of COVID-19-associated hospital admissions in residents from participating care homes to the COVID-19 Datastore. Linkage to individual pseudonymised IDs will allow comparison to the routinely collected hospital admission data once available.

To inform estimates of incidence rates which form the primary and secondary outcomes, we will collect the total number of residents in the home on a weekly basis from providers as this will allow us to estimate the denominator. We will also collect the total number of staff on a weekly basis from each home to inform estimates of testing uptake and explore the feasibility of collecting data on the number of staff who opt-out of asymptomatic testing (in the intervention arm).

Care home level (aggregate) data will be collected from providers on dates of care home closures, use of disease control measures, staff sickness absence and employment of agency staff to inform health economic analyses. We will explore the feasibility of collecting care home level data on fees paid by residents who are funded by the local authority, and whether it is feasible to collect more detailed information on healthcare utilisation, such as primary care consultations and use of antivirals in residents.

Data on outbreak events (dates, size) will be obtained from the UK Health Security Agency (UKHSA) Adult Social Care Team. Data on the local incidence of COVID-19 and cocirculation of other respiratory viruses will be obtained from the UKHSA and/or the Office of National Statistics Covid Infection Survey.

## Data management

Individual-level trial data will be stored in the UCL Data Safe Haven,[25] which is hosted by UCL. All identifiable data will be held only by individual care homes or NHS England (NHSE), who will act as data processor on behalf of UCL. These databases are protected by multilayer firewalls with full data encryption at rest and in transit.

For qualitative interviews, data collection will occur remotely using secure communication methods and be conducted by University of Strathclyde (UoS) researchers.

On completion of transcription at UoS, the pseudoanonymised data will be stored on the UoS network in a secure, restricted access folder for 5 years from the time of end of trial. Raw data will be destroyed once transcription and quality checks have been performed. Consent forms obtained via interviews and focus groups in the process evaluation will also be stored securely.

## Statistical analysis

Analysis of the primary outcome, and secondary outcomes expressed as event incidence, will be based on Poisson or negative binomial regression with cluster-robust SEs, adjusting for calendar time and key care home characteristics used in the restricted randomisation such as provider, region and size. We will also explore whether the intervention effect differs according to care home size, and other characteristics such as proportion of temporary staff. Unadjusted effect estimates from these analyses will be reported for completeness. Using interaction terms, we will explore whether the effect of the intervention on the primary outcome differed between time periods defined by the national recommendations for testing in the routine care arm, should these change during data collection.

Analysis of the primary outcome will include all trial care homes (intention-to-treat analysis), and so represent a treatment policy estimand. We will also define an implementation score based on the frequency and proportion of staff testing at each home based on data the homes provide, which may vary over time. As an exploratory analysis we will assess whether the primary outcome is associated with this implementation score within the intervention arm and express the effect of the intervention relative to control arm for different levels of implementation. This analysis will be based on the same regression method as used for the primary analysis.

## Health economic analysis

The health economic analysis will investigate the cost-effectiveness and cost consequences of the testing intervention taking an NHS, personal social services and a societal perspective using a lifetime horizon (according to care home resident average age and life expectancy). The within-trial costs and outcomes in intervention and control groups will be examined from each perspective. Cost-effectiveness of the intervention in terms of the primary outcome and in terms of all-cause mortality will also be examined. Costs of admission will be excluded from the total costs under consideration in this case.

We will also examine cost-effectiveness in terms of the secondary outcomes of cases prevented and resident deaths prevented, and outcomes of hospital admission and number of outbreaks alongside costs offset/additional costs incurred in a cost–consequences analysis.

## Process evaluation

There is major diversity across care homes, for example, in terms of provision of care, resident population, care

home size and the care home workforce. As a result, it is essential to consider the feasibility and sustainability of the intervention and how contextual factors might impact on the ability to scale it, if the trial suggests it is effective and cost-effective. These issues will be addressed in the process evaluation, which aims to understand intervention roll out and identify areas for optimisation to inform future intervention scale-up, should the testing approach prove effective and cost-effective.

The objectives of the process evaluation are as follows:

► To determine intervention acceptability.
► To determine the role of context in shaping the way, the intervention operated.
► To determine what can be learned about intervention fidelity and adaptation.
► To determine which intervention components worked as anticipated and which need further modification.
► To investigate unanticipated intervention effects.
► To determine what can be learned from the control group.

The process evaluation will develop implementation guidance and training packages ready for future scale up as well as details of minimal care home requirements and staff competencies necessary for intervention delivery.

Qualitative data will be collected from 28 (10%) care homes evenly distributed across each intervention and control arms and spaced across time.

### Patient and public involvement

Patient and public involvement (PPI) has already informed the development of this programme, by highlighting the barriers to testing and the need to capture its adverse impacts on staff, residents and providers. Public advisors have also emphasised the importance of developing a strong plan for implementation, recognising the financial implications of long-term use of testing and sickness payments, informing our emphasis on implementation in WP5.

The PPI team will deliver the following objectives:

► To ensure that the 'voice and views' of the public regarding regular testing for COVID-19 are heard by the research team and the wider stakeholder group.
► To create an open, inclusive culture enabling effective communication between the study team, PPI group and the wider stakeholder and oversight groups.
► To agree an approach to communicate outputs from the trial to different audiences, including care home staff, residents and their families and the public using a variety of media.

### ETHICS AND DISSEMINATION
### Study monitoring

An independent trial steering committee (TSC) and data monitoring and ethics committee (DMEC) will be formally responsible for programme oversight, ensuring the study is conducted in compliance with regulations. The DMEC will also be responsible for monitoring the

accumulating data and making recommendations to the TSC on whether the trial should continue as planned.

A trial management group will be responsible for the design, coordination and strategic management of the trial.

### Safety reporting

Staff at the sites randomised to asymptomatic testing will report the occurrence of serious adverse events (SAEs) considered 'related' to the intervention only. In such cases, site personnel will complete an SAE report within 24 hours of notification of the event. Clinical review of any SAEs will take place and be reported to the Research Ethics Committee if deemed both 'related' to the trial intervention and 'unexpected' in line with UK Health Research Authority (HRA) non-drug trial reporting requirements.

### Research ethics approval

The study has been approved by the London—Bromley Research Ethics Committee (reference number 22/LO/0846) and the HRA (22/CAG/0165).

### Consent and opt-out

Care home providers and home managers will be asked if their care home(s) are willing to participate in the trial. High staff turnover, in conjunction with the large number of care homes participating in the trial, means that it is not feasible to obtain individual consent from staff or regarding the use of testing data. Staff and residents have the option of opting out from the processing and analysis of their individual-level data within this study at any time during the study.

Identifiable data submitted by care homes as part of the study will be pseudonymised by NHSE before it is provided to the research team. This study has section 251 support to allow the disclosure of confidential patient information (regarding testing in staff) from care homes to NHSE, for the purposes of monitoring uptake of the testing in the control and intervention arms of the trial.

The study will also collect limited individual-level identifiable data from residents to ensure the primary outcome can be determined accurately. It is not feasible to seek individual-level consent from every resident for the use of these data due to high levels of cognitive impairment in residents and excluding data from a large proportion of residents would compromise the scientific value of the trial and the subsequent generalisability of trial findings. This study has section 251 support to allow the disclosure of confidential patient information (regarding residents admitted to hospital) from care homes to NHSE, for the purposes of linkage to the COVID-19 datastore and to enable NHSE to use confidential patient information from SARS-CoV-2 tests to link to other NHS datasets within the COVID-19 datastore.

Care home managers in the subset of homes selected for qualitative data collection (focus groups or one to one interviews) will be asked to disseminate recruitment

materials to staff within the home via word of mouth, email, or other routine modes of communication. On receipt of staff contact details, interested staff will then be sent participant information sheets about the study, given the option to ask questions about the study, complete on-line consent forms and provide brief sociodemographic details to enable the study team to monitor total sample composition. Having checked on-line consent has already been given and after exploring any remaining unanswered questions raised by the Participant Information Sheets (PIS), the participants will be asked to also give recorded oral consent to participate.

### Confidentiality

All data will be handled in accordance with the Data Protection Act 2018, the UK General Data Protection Regulation and subsequent updates and amendments.

### Dissemination policy

The results of the trial will be disseminated regardless of the direction of effect. The publication of the results will comply with a trial-specific publication policy and will include submission to open access journals.

A lay summary of the results will also be produced to disseminate the results to participants. A summary of results will be included online in the publicly accessible HRA website within 12 months of the date of trial closure. A statistical analysis plan will also be published under open access arrangements.

## DISCUSSION

The proposed research faces several specific methodological and operational challenges, primarily because the study is being conducted at pace in a dynamic epidemiological and policy context. We outline the main challenges and potential mitigations here.

In designing this trial, we worked closely with care home staff and providers to design a testing intervention that would be feasible and acceptable. In particular, we worked closely with each care provider participating in the study to understand how they process and organise sick pay and employ agency staff and the likely costs. We met with the providers' Human Resources (HR) teams and also members of their senior management team. We established a flexible approach to reimbursing sick pay and agency staff to ensure all legitimate costs will be covered. Providers will be asked to provide evidence of the actual costs accrued each month and this will be verified by the funder before payments are released, ensuring that compensation for staff sickness is adequate.

We have also organised a series of stakeholder engagement events with diverse care home staff and representatives from the care home sector to agree the content of the intervention, which has been designed to maintain compliance with testing and ensure staff in intervention care homes will not be disadvantaged. Engagement events have included consideration of testing acceptability, how to increase uptake of testing and logistics related to support payments, such as when and how they should be paid to staff. These events will also provide an opportunity to discuss concerns related to potential consequences for staff and organisations as a result of participating in the trial. All these issues were taken into account when designing the testing protocol and associated support payments. Care home staff in participating care homes were noted to be extremely familiar with the process of regular asymptomatic testing and reimbursement of sick pay as this was in place throughout the pandemic in England.

There is a significant question over whether the trial will be sufficiently powered to detect a statistically significant outcome. This is predicated on both the epidemiological event rate during the intervention period, and willingness of sufficient numbers of care homes to participate.

We are acutely aware that there is a risk that rates of COVID-19 might decline from estimates used from previous years (making it impossible to achieve significance for the primary outcome), but we have taken the view that this is unique opportunity to try to generate data on the effectiveness, benefits and harms of regularly testing asymptomatic staff for COVID-19 (with financial support for staff sickness and agency staff backfill) to prevent severe outcomes in residents, which would be lost if we did not attempt this trial. At the time of trial design and the application for funding (August 2022), it was very unclear whether there would be a resurgence of COVID-19 in autumn/winter associated with the emergence of a new variant.

In the event that the trial's primary outcome cannot be delivered as planned, the non-trial WPs will still generate valuable evidence to inform future use of testing in care homes, by characterising barriers and facilitators to testing, estimating costs of the testing intervention and generating models that can be used to estimate the impact of testing on infections and hospital admissions under different epidemiological scenarios.

While we considered including testing for visitors as part of the intervention, this would have introduced further complexity regarding consent. It would also have strayed from the approach that was adopted during the pandemic in England, which is what we wanted to evaluate.

We note that results of a trial investigating the benefits of COVID-19-related staff testing and sickness support payments in a care home context would have been beneficial earlier in the course of the pandemic. The challenge in trying to do this has been that it would have been extremely difficult to persuade public health agencies (in the UK or elsewhere) that it was reasonable to withhold regular asymptomatic testing in control homes when there were high levels of COVID-19 transmission in the community. We feel it remains a highly salient research question and the results should help inform policy for care home preparedness, both in relation to COVID-19 but also in support of future research into wider institutional infectious disease transmission mitigation.

While there are significant methodological challenges to conducting this study, it is our view that we need to learn how to do trials at pace and scale in care homes, to improve the quality of care for residents. In addition to generating important evidence on the effectiveness, benefits and harms of asymptomatic testing for COVID-19 in staff to prevent severe outcomes in residents, this trial will provide important learning to inform the design and delivery of future care home trials.

### Trial registration and reporting guidelines

The VIVALDI-CT was registered with the International Standard Randomised Controlled Trial Number website (ISRCTN 13296529)[26] on 5 December 2022 and the protocol adheres to the Standard Protocol Items: Recommendations for Interventional Trials (SPIRIT) 2013 statement.[27]

**Author affiliations**
[1]Institute of Health informatics, University College London, London, UK
[2]Institute for Global Health, University College London, London, UK
[3]Comprehensive Clinical Trials Unit, Institute of Clinical Trials and Methodology, University College London, London, UK
[4]Department of Primary Care and Public Health, Brighton and Sussex Medical School, Brighton, UK
[5]UK Health Security Agency, London, UK
[6]Centre for Dementia Studies, Department of Neuroscience, Brighton and Sussex Medical School, Brighton, UK
[7]Care Policy and Evaluation Centre, The London School of Economics and Political Science, London, UK
[8]Department of Infectious Disease Epidemiology, London School of Hygiene & Tropical Medicine, London, UK
[9]Department of Psychological Sciences and Health, University of Strathclyde, Glasgow, UK
[10]Division of Population Health, Health Services Research & Primary Care, School of Health Sciences & Manchester Academic Health Science Centre, The University of Manchester, Manchester, UK
[11]Manchester Academic Health Science Centre, The University of Manchester, Manchester, UK
[12]Academic Unit of Injury, Recovery and Inflammation Sciences (IRIS), School of Medicine, University of Nottingham, Nottingham, UK
[13]Applied Research Collaboration-East Midlands (ARC-EM), NIHR, Nottingham, UK

**Acknowledgements** We are grateful to the care homes who have already agreed to participate in the VIVALDI-CT. We would also like to acknowledge the role of the care homes participating in the VIVALDI Study for sharing their learning from the pandemic with us. We also acknowledge the support of the independent members of the Trial Steering Committee (TSC): Professor Alistair Hay (University of Bristol). Jennifer Thompson (LSHTM). Michael Larkin (Aston University). Zoe Fry (Outstanding Society). Samantha Crawley (Bracebridge). Margaret Ogden (independent PPI member). As well as the Data Monitoring and Ethics Committee (DMEC):Karla Hemming (University of Birmingham). Tania Kalsi (Guy's and St Thomas' NHS Foundation Trust). Terry Quinn (University of Glasgow). We would also like to thank members of the UKHSA:Tom Fowler (UKHSA). Alex Barton (DHSC). Jackie Cassell (UKHSA). Sarah Tunkel (UKHSA).

**Contributors** NA, JB, MKr, LS and PF developed the study protocol and contributed to the writing of the manuscript. OS and AC were involved in development of the statistical analysis facets of the trial and contributed to the writing of the manuscript. CH and MKn were involved in development of the health economic analysis facets of the study and contributed to the writing of the manuscript. RL was involved in development of the process evaluation of the study and contributed to the writing of the manuscript. DC and JAC were involved in development of the health-related quality of life facets of the study and contributed to the writing of the manuscript. LG was involved in development of the economic modelling of the study and contributed to the writing of the manuscript. IC-S is the trial manager for

the study. RF will oversee data collection and management. MR, AV, AG, NF and SH are coapplicants responsible for supporting the operationalisation of the study. LS and PF are the co-chief investigators for the study. All authors critically reviewed and approved the final version.

**Funding** This work is supported by the NIHR Health and Social Care Delivery Research (HSDR) Programme number (154310). Costs associated with SARS-CoV-2 testing including support payments for care home staff and for care homes to fund agency staff backfill will be funded by the UK Health Security Agency (UKHSA). The VIVALDI-CT is sponsored by University College London, represented by the UCL Comprehensive Clinical Trials Unit (UCL CCTU).

**Competing interests** None declared.

**Patient and public involvement** Patients and/or the public were involved in the design, or conduct, or reporting, or dissemination plans of this research. Refer to the Methods section for further details.

**Patient consent for publication** Not applicable.

**Provenance and peer review** Not commissioned; externally peer reviewed.

**ORCID iDs**
Natalie Adams http://orcid.org/0000-0002-6131-480X
James Blackstone http://orcid.org/0000-0003-4335-5269
Jackie A Cassell http://orcid.org/0000-0003-0777-0385
Adam Gordon http://orcid.org/0000-0003-1676-9853
Andrew Copas http://orcid.org/0000-0001-8968-5963
Paul Flowers http://orcid.org/0000-0001-6239-5616

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
