## [Reviewer comments · BMJ Open]

ARTICLE DETAILS

TITLE (PROVISIONAL)	Shaping care home COVID-19 testing policy: A Protocol for a pragmatic cluster randomised controlled trial of asymptomatic testing compared to standard care in care home staff (VIVALDI-CT).
AUTHORS	Adams, Natalie; Stirrup, Oliver; Blackstone, James; Krutikov, Maria; Cassell, Jackie; Cadar, Dorina; Henderson, Catherine; Knapp, Martin; Goscé, Lara; Leiser, Ruth; Regan, Martyn; Cullen-Stephenson, Iona; Fenner, Robert; Verma, Arpana; Gordon, Adam; Hopkins, Susan; Copas, Andrew; Freemantle, Nick; Flowers, Paul; Shallcross, Laura

VERSION 1 – REVIEW

REVIEWER	Ismaila, Hamza Ghana Health Service, Office of Director-General
REVIEW RETURNED	10-Jul-2023

GENERAL COMMENTS	Some minors revisions have been proposed. (The reviewer provided a marked copy with additional comments. Please contact the publisher for full details.)
---

REVIEWER	Chevalier, Judith A. Yale University
REVIEW RETURNED	18-Jul-2023

GENERAL COMMENTS	This protocol is well-explained. And it seems feasible. There are repeated mentions of staff consent and staff opt-out to the routine testing, and there is funding to compensate homes who replace staff who take sick days and sick days are paid. However, it is not at all clear to me the extent to which staff have been shielded from negative employment consequences of testing positive in performance evaluations, whether the staff compensation for sick days are adequate, whether sick days come from an annual bank that are finite in number, and what the consent materials say about the consequences of testing positive. (So I don't feel qualified to answer the question of whether the consent materials are adequate.) My second concern stems from the power calculations. I would predict that it is extremely unlikely that this protocol could produce any statistically significant efficacy of workplace testing. One: it excludes non-employee care providers who visit the home (GPs, etc). There are plenty of opportunities for COVID to transmit from the community to the care home among untreated workers,
--

	visitors, patient transfers, and workers who opt out. The power calculations are contemplating COVID rates similar to 2021/2022 and (implicitly) hospital rates conditional on infection that are similar over time. Given vaccination and natural immunity, this seems unlikely and my strong suspicion is that the study is underpowered to detect differences in resident covid hospitalization rates.
--	---

REVIEWER	Wasserman, Michael California Association of Long Term Care Medicine
REVIEW RETURNED	05-Sep-2023

GENERAL COMMENTS	In many ways, this research is groundbreaking. My only criticism is that it wasn't performed 2 years earlier. The field of geriatric medicine desperately needs research like this in the nursing home setting in order to better inform
--

VERSION 1 – AUTHOR RESPONSE

Reviewer: 1

Mr. Hamza Ismaila, Ghana Health Service

Comments to the Author:

- Some minors revisions have been proposed. See file attached

Thank you for your comments, we have addressed these via tracked changes directly in the revised manuscript

Reviewer: 2

Prof. Judith A. Chevalier, Yale University

Comments to the Author:

- This protocol is well-explained. And it seems feasible.

Thank you for your feedback.

There are repeated mentions of staff consent and staff opt-out to the routine testing, and there is funding to compensate homes who replace staff who take sick days and sick days are paid. However, it is not at all clear to me the extent to which staff have been shielded from negative employment consequences of testing positive in performance evaluations, whether the staff compensation for sick days are adequate, whether sick days come from an annual bank that are finite in number, and what the consent materials say about the consequences of testing positive. (So I don't feel qualified to answer the question of whether the consent materials are adequate.)

Thank you. For this component of the intervention to work we had to work closely with each of the care providers who were participating in the study to understand how they process and organise sick pay and employ agency staff, and the likely costs. We met with the providers HR teams and also met with members of their senior management team. We had a flexible approach to reimbursing sick pay and agency staff to ensure all legitimate costs were covered. Providers were asked to provide evidence of the actual costs accrued each month (they supplied de-identified staff rotas) and this information was verified by the funder before payments were released. This ensured that compensation for staff sickness was adequate.

As part of the design of this trial, we organised a series of stakeholder engagement events with diverse care home staff and representatives from the care home sector to agree the content of the intervention, which was designed to maintain compliance with testing and to ensure that staff in

intervention care homes were not disadvantaged. The engagement events included consideration of testing acceptability and self-isolation support payments. It also provided an opportunity to discuss concerns related to potential consequences on staff and organisations as a result of participating in the trial. These considerations were taken into account when designing the testing protocol and associated support payments. It is important to note that care home staff in participating care homes were extremely familiar with the process of regular asymptomatic testing and reimbursement of sick pay as this was in place throughout the pandemic in England.

- My second concern stems from the power calculations. I would predict that it is extremely unlikely that this protocol could produce any statistically significant efficacy of workplace testing. One: it excludes non-employee care providers who visit the home (GPs, etc). There are plenty of opportunities for COVID to transmit from the community to the care home among untreated workers, visitors, patient transfers, and workers who opt out. The power calculations are contemplating COVID rates similar to 2021/2022 and (implicitly) hospital rates conditional on infection that are similar over time. Given vaccination and natural immunity, this seems unlikely and my strong suspicion is that the study is underpowered to detect differences in resident covid hospitalization rates.

Thank you for your comments. When we designed the trial in summer 2022 it was unclear if there would be an additional wave of COVID-19 in winter 2022, and the emergence of new SARS-CoV-2 variants remained a real concern. We were acutely aware that there was a risk that rates of COVID-19 might decline (making it impossible to achieve the primary outcome), but we took the view that there was a unique opportunity to try to generate data on the effectiveness, benefits and harms of regularly testing asymptomatic staff for COVID-19 (with financial support for staff sickness and agency staff backfill) to prevent severe outcomes in residents, which would be lost if we did not attempt this trial. Even if the trial cannot be delivered as planned, the non-trial work packages will generate valuable evidence to inform future use of testing in care homes, by characterising barriers and facilitators to testing, estimating costs of the testing intervention and generating models that can be used to estimate the impact of testing on infections and hospital admissions under different epidemiological scenarios.

Whilst we considered including testing for visitors as part of the intervention, this would have introduced further complexity regarding consent. It would also have strayed from the approach that was adopted during the pandemic, which is what we wanted to evaluate.

Reviewer: 3

Dr. Michael Wasserman, California Association of Long Term Care Medicine

Comments to the Author:

- In many ways, this research is groundbreaking. My only criticism is that it wasn't performed 2 years earlier. The field of geriatric medicine desperately needs research like this in the nursing home setting in order to better inform

Thank you for your comments. We completely agree with the reviewer on this point that it would have been better to perform this trial earlier in the pandemic. However, the challenge in trying to do this is that it would have been extremely difficult to persuade public health agencies that it was reasonable to withhold regular asymptomatic testing in control homes when there were high levels of COVID-19 transmission in the community.